# Navigating the Latest Hepatitis B Virus Reactivation Guidelines

**DOI:** 10.3390/diseases13110355

**Published:** 2025-11-01

**Authors:** Zeyad Elharabi, Jowana Saba, Hakan Akin

**Affiliations:** Division of Gastroenterology, Department of Internal Medicine, Texas Tech University Health Sciences Center, Lubbock, TX 79430, USA; zeyad.elharabi@ttuhsc.edu (Z.E.); jowana.saba@ttuhsc.edu (J.S.)

**Keywords:** hepatitis B reactivation, guidelines, immunosuppressive therapy, risk stratification, antiviral

## Abstract

Hepatitis B virus (HBV) infection is a global health concern with an estimated 254 million people with chronic HBV infection. The utilization of immunosuppressive therapies (ISTs) is increasing and expanding continuously with new agents being implemented across multiple medical disciplines. The occurrence of HBV reactivation (HBVr) during or after IST varies from 15% to 50% in HBsAg-positive individuals and can be higher than 75% after stem cell transplantation. HBVr is gaining increasing significance in contemporary clinical practice. The American Gastroenterological Association (AGA) in 2025, the European Association for the Study of the Liver (EASL) in 2025, and the Asian Pacific Association for the Study of the Liver (APASL) in 2021, published their most recent clinical guidelines as major societies in the area, which enables us to better predict and manage HBVr. This narrative review focuses on comparing these three current guidelines, highlighting key similarities and differences to provide valuable guidance for practitioners navigating the complex, sometimes conflicting recommendations, thereby aiding clinicians in their decision-making. The risk of HBVr during IST has been stratified into three categories in all three guidelines: high (>10%), moderate (1–10%), and low (<1%). The effectiveness of prophylaxis scales with baseline risk for HBV reactivation. Prophylaxis is clearly cost-effective for high-risk patients, potentially beneficial for those at moderate risk, and generally may not be justified for low-risk individuals. Entecavir (ETV), tenofovir disoproxil fumarate (TDF), and tenofovir alafenamide (TAF) are all highly effective in preventing HBV reactivation during immunosuppression and all are considered to be economically viable options for HBVr high risk patients. When selecting among these agents, safety considerations—particularly renal and bone toxicity—and insurance coverage remain the primary factors directing clinical decision-making.

## 1. Introduction

Hepatitis B virus (HBV) infection is a global health concern associated with significant hepatic morbidity and mortality, including complications such as cirrhosis and hepatocellular carcinoma. As of 2022, an estimated 254 million people live with chronic HBV infection [1]. HBV-related cirrhosis and chronic liver disease accounted for an estimated 1.08 million deaths worldwide. Only 13% of people living with chronic hepatitis B infection had been diagnosed, and only 3% received antiviral therapy at the end of 2022 [2]. The burden in low and middle incomes countries is even higher, especially in sub-Saharan Africa and the Western Pacific. For example, in South Sudan the prevalence rate is higher than 20%. This might be explained by difficulties in vaccination, diagnosis, and obtaining therapy [3,4].

Especially in the developed countries of the world, where IST is increasingly used, besides cirrhosis and liver cancer, as well-known HBV-related deaths, acute flares and viral reactivation further contribute to the disease burden. HBVr refers to increased viral replication from a previously suppressed or dormant state, often resulting in higher viral loads and hepatitis [5]. This can occur in both HBsAg-positive individuals and those who are HBsAg-negative/anti-HBc-positive, due to the persistence of covalently closed circular DNA (cccDNA), particularly during IST [6].

The clinical importance of HBVr was first documented in the oncology context, initially observed among lymphoma patients receiving chemotherapy during the 1970s and 1980s [7,8]. It has since been reported in other settings involving immunosuppression, such as organ transplantation, nephrology, rheumatology, or gastroenterology treatments. Acute HBV flares or reactivation can occur spontaneously or due to various triggers. Key causes include ISTs (such as chemotherapy, anti-rejection drugs, B cell-depleting agents, corticosteroids, biologics, antimetabolites, tyrosine kinase inhibitors, and immune checkpoint inhibitors). Reactivation may also be linked to changes in antiviral therapy, drug resistance, or coinfections with other liver viruses (hepatitis D, C, A, and E). Other factors include coexisting conditions such as human immunodeficiency virus (HIV), tuberculosis, pregnancy, ESRD (end stage renal disease), major post-bariatric weight loss, or hepatitis C eradication [9]. The occurrence of HBVr during or after IST varies from 15% to 50% in HBsAg-positive individuals and can be higher than 75% after stem cell transplantation. For HBsAg-negative/anti-HBc-positive individuals, HBVr is less frequent but may exceed 10% in specific circumstances, such as with B cell-depleting therapies [10,11].

The severity of HBV flares ranges from mild transaminase elevations to acute liver failure, influenced by factors such as hepatic reserve, baseline virologic status, and the extent of immunosuppression [12]. According to the AGA guidelines, a hepatitis flare due to HBVr is characterized by a serum alanine aminotransferase level at least threefold greater than baseline and exceeding the reference range [13,14]. “Severe” reactivation hepatitis may be indicated by ALT elevations greater than ten times the upper limit of normal or by clinical signs of liver failure, such as jaundice and coagulopathy [12].

The host’s immune system primarily suppresses the HBV infection. The HBVr is triggered by the disruption of immune surveillance. If the balance between host immune systems and viral replication distorted, unchecked HBV replication ensues; subsequently, restoration of immunity may precipitate immune-mediated liver injury [15,16].

The immunological mechanisms underlying the prevention of hepatitis B virus reactivation (HBVr) are multifaceted and remain to be explored further. Host defense organization against HBVr involves a coordinated interplay of immune cells—including natural killer (NK) cells, CD8^+^ cytotoxic T lymphocytes, CD4^+^ helper T cells, B lymphocytes, and other innate immune components—alongside the dysregulated cytokine networks such as interleukin IL-12, IL-18, IL-1β, IL-6, and tumor necrosis factor-alpha (TNF-α) [17,18]. Additionally, immunological checkpoints and signaling regulatory pathways contribute to immune modulation, notably by downregulating activating NK cell receptors (e.g., NKp30, NKp46, and CD56dim), thereby impairing antiviral cytotoxicity and facilitating viral persistence [19].

Ongoing research continues to elucidate the mechanisms underlying HBVr, particularly in the context of expanding immunosuppressive and anticancer therapeutic regimens. Recent findings have identified a novel molecular pathway by which paclitaxel (PTX) induces HBV reactivation. This phenomenon appears to result of direct activation of HBV replication through upregulation of HBV core promoter activity by the transcription factor activator protein-1 (AP-1), which is a central regulatory node of several immune-related signaling pathways, including IL-17, NF-κB, and MAPK [20].

Prophylactic antiviral therapy is effective in preventing reactivation [21,22,23]. Without timely recognition and management, reactivation can lead to a severe, potentially fatal outcome [24].

Thorough risk assessment, comprehensive virologic screening prior to immunosuppression, prompt initiation of antiviral prophylaxis, and continuous monitoring are advised. Recent clinical guidelines enable healthcare professionals to better predict and manage HBVr. However, medical guidelines may not cover every scenario, as clinical decisions often involve judgment and the application of medical knowledge to individual preferences and circumstances. This review summarizes and compares various approaches for HBVr diagnosis and management based on the latest guidelines of three major medical associations.

## 2. Materials and Methods

This study utilized a narrative review methodology to examine the latest guidelines about emerging screening, prophylaxis, and management recommendations concerning Hepatitis B virus reactivation (HBVr). The narrative review format was chosen for its flexibility and interpretive capacity, allowing for a comparative assessment of recommendations issued by diverse medical societies across geographic and disciplinary boundaries. Relevant literature was identified through a non-systematic search of electronic databases—PubMed, Scopus, and Web of Science—covering publications from January 2000 to May 2025. A broad range of keywords and Medical Subject Headings (MeSH) terms were employed, including Hepatitis B, Latent Infection, Guideline [Publication Type], Practice Guideline [Publication Type], Consensus, Re-view [Publication Type], Systematic Review [Publication Type], Meta-Analysis [Publication Type], and Expert Opinion (Expert Testimony), used in various combinations to capture a comprehensive spectrum of HBVr-related publications. Additional sources were identified through manual screening of reference lists and citation tracking. Inclusion criteria were limited to peer-reviewed articles. Exclusion criteria comprised non-English publications, abstracts without full-text access, and studies deemed irrelevant to the scope of the review. Consistent with the narrative review approach, no formal quality scoring or meta-analysis was performed. The selected literature was critically evaluated for its conceptual relevance, methodology, and contribution to the evolving understanding of HBVr management.

## 3. Definition of Hepatitis B Reactivation (According to the Major Gastroenterology/Liver Societies)

The American Gastroenterological Association (AGA) defines HBVr as a loss of immune control in patients positive for HBsAg or anti-HBc. According to the 2015 and 2025 AGA guidelines, HBVr is either a new appearance of HBV-DNA in previously undetectable patients or a tenfold increase from baseline; detection of HBsAg or hepatitis B e antigen may also serve as surrogates. The AGA accepts variations in these criteria, such as an HBV-DNA rise to 2000 IU/mL and serum alanine aminotransferase (ALT) above 100 IU/L, in line with clinical practice [13,14].

The EASL (European Association for the Study of the Liver) 2025 Hepatitis B guideline defines HBVr as a sudden escalation in HBV replication occurring in HBsAg-positive individuals with an inactive profile (HBeAg-negative chronic infection), or in HBsAg-negative individuals with resolved hepatitis B (HBsAg-negative, anti-HBc-positive). HBVr is identified by the re-emergence of HBV DNA levels (>100 IU/mL) or HBsAg seroreversion in individuals previously exhibiting undetectable markers, or by at least a tenfold increase in HBV DNA from baseline [25].

APASL (Asian Pacific Association for the Study of the Liver) defines HBVr in two scenarios in the 2021 guidelines. First, chronic HBV infection (HBsAg positive) is exacerbated if there is a ≥2 log increase in HBV DNA from baseline, or detection of HBV DNA > 100 IU/mL after previously being undetectable at baseline. Second, past HBV infection (HBsAg-negative, anti-HBc positive) is defined as reactivated after IST if HBsAg seroconverts from negative to positive, or HBV DNA becomes detectable without HBsAg [26].

## 4. Risk Factors for Hepatitis B Reactivation

### 4.1. Viral Factors

The risk of reactivation differs in HBsAg positive patients compared to the HBsAg negative/anti-HBc positive patients, and is estimated to be five to eight times higher; therefore, prophylaxis therapy recommendations differ for both groups [14,26,27]. In patients who are HBsAg positive, the risk increases with the increase in the HBV DNA level, and it is also affected by the HBeAg seropositivity [26]. The EASL guidelines, in contrast to the AGA and APASL guidelines, use the HBV DNA level when classifying the reactivation risk of ISTs [14,25,26]. In patients who are HBsAg negative/anti-HBc positive, unlike in HBsAg positive patients, anti-HBs antibodies might be protective against reactivation, but the data is limited [14,26]. Therefore, the AGA and APASL guidelines agree against the use of anti-HBs antibodies for risk stratification [14,26]. There is no clear link between the HBV genome sequence and the risk of reactivation [26].

### 4.2. Host Factors

#### 4.2.1. Demographics

Male sex and older age increase the risk of reactivation [25,26,28].

#### 4.2.2. Underlying Disease

The risk of reactivation is also affected by the presence of other co-existent viral infections (HIV, HCV, or HDV) [26].

Hepatitis C virus (HCV) treatment by direct-acting antivirals (DAAs) can lead to reactivation of HBV infection, whether during or after completion of the DAA therapy, and whether or not the HBV DNA is detectable. However, the risk of reactivation may be higher in patients with detectable HBV DNA levels [26,27]. Furthermore, presence of cirrhosis in patients treated with DAAs leads to a higher risk of reactivation [26,29].

The type of disease treated by ISTs can also affect the risk of reactivation [25,26].

#### 4.2.3. Immunosuppressive Therapy

Patients on medications that can cause immunosuppression and that are used to treat cancer or a variety of immunologic diseases are at higher risk of HBVr. Patients that are receiving immunosuppression due to undergoing transplantation are also at increased risk [28]. The risk of reactivation is affected by the type of immunosuppressive regimen, the duration of therapy, as well as the disease in which the immunosuppression is used. [25,26]. The risk is considerably higher with B cell depleting agents (e.g., rituximab and CAR-T therapy) [14,26]. While some of the recommendations regarding the immunosuppressive regimen given in the guidelines covered in this article are based on studies, others were based on expert opinion and extrapolation due to the lack of relevant studies.

## 5. Prevention of HBV Reactivation

### 5.1. Screening

While the USPTF, in their 2020 guidelines, limit their recommendations for HBV screening in the general population to adolescents and adults at increased risk of HBV infection [30]; the CDC, in their March 2023 recommendations, recommend screening all adults 18 years or older for HBV using HBsAg, anti-HBs antibodies, and total anti-HBc antibodies [29].

The AGA, APASL, AASLD, and EASL in their latest guidelines recommend testing all patients who are at possible risk of HBVr for HBV. The American Society of Clinical Oncology (ASCO) guidelines recommend HBV testing for all patients who are planned to receive anticancer therapy. The APASL and ASCO guidelines recommend testing for HBsAg, anti-HBs antibodies, and anti-HBc antibodies as part of the initial testing [26,31]. The EASL guidelines recommend testing all patients planning to undergo IST for HBsAg and anti-HBc antibodies. If they are HBsAg negative/anti-HBc positive, they should be tested for HBV DNA to check for active replication, and this is used in the risk assessment. In those patients, they also suggested checking anti-HBs and anti-HBc antibody levels as it aids in the risk assessment. Higher levels of anti-HBc antibodies have been shown to correlate with increased reactivation risk, whereas higher levels of anti-HBs antibodies (≥100 IU/L) correlate with decreased risk. Care should be taken in patients already undergoing immunosuppression as the antibody levels can be lower because of the IST [25]. AASLD recommends screening with both HBsAg and anti-HBc antibodies, but it does not find a clear role for anti-HBs antibodies [32].

For HBsAg-positive patients, the APASL guidelines recommend added testing for serum HBV DNA and possibly serum quantitative hepatitis B surface antigen (qHBsAg). However, they do not recommend testing for anti-HBs antibody titers, anti-HBc titers, HBV core-related antigen, ultra-sensitive HBsAg, and HBV RNA due to the lack of sufficient evidence to guide their use [26].

The AGA guidelines, recommend testing at least for HBsAg and anti-HBc antibodies, and HBV DNA testing should be done if either the HBsAg or the anti-HBc antibodies are positive [14].

The EASL and ASCO guidelines recommend HIV testing for all patients that will receive entecavir, tenofovir disoproxil fumarate (TDF), or tenofovir alafenamide fumarate (TAF) due to their potential to cause mutations leading to HIV resistance [27,31].

### 5.2. Vaccination

Patients who are found to be negative for HBV during screening may be considered for HBV vaccination. Studies are needed to investigate whether vaccinating patients with isolated anti-HBc antibodies to achieve positive anti-HBsAg antibodies is beneficial and protective against reactivation [31].

### 5.3. Risk Stratification

The AGA risk classification for HBVr published in 2014 and continued in their Feb 2025 guidelines, classified the risk into three groups: high (>10%), moderate (1–10%), and low (<1%). This was also adopted by the EASL guidelines; however, there are differences in the placement of patients in the categories (see Table 1) [14,25,27]. The APASL guidelines use the same categories for classifying the risk but also differ in how they place patients in each category [26]. The ASCO guidelines do not adopt this categorization due to the lack of data that reliably quantifies the HBV risk, but they do agree that there is a well-known high risk of reactivation with anti-CD20 monoclonal antibodies and stem cell transplantation [31].

Classification into the different categories depends on the type and intensity of therapy used and the HBV serological markers. EASL differs from AGA and APASL in adding HBV DNA level as a factor in in their risk assessment [25].

The EASL 2025 guidelines were cautious in their approach and preferred classifying patients in the higher risk group when the evidence was unclear [25].

#### 5.3.1. High Risk Category

##### HBsAg Positive Patients (in the EASL Guidelines Also Applies to Patients That Are HBsAg Negative/Anti-HBc Positive with Positive HBV DNA)

The AGA, APASL, and EASL guidelines agree that patients who are on B cell-depleting agents (for example, anti-CD20 monoclonal antibodies) are at high risk of reactivation whether they are HBsAg-positive or HBsAg negative/anti-HBc positive [14,25,26].

According to the AGA and EASL guidelines, this category includes patients on therapy with anthracyclines, anti-TNF agents, tyrosine kinase inhibitors, CAR-T cell therapy, or JAK inhibitors [14,25]. The APASL guidelines are similar but differ in that they did not include CAR-T cell therapy and were uncertain about anti-IL6 therapy. They also exclude the lower potency anti-TNF agent etanercept from this category [26].

In the AGA guidelines, this category also includes patients who are co-infected with HCV, and receiving DAA regardless of the HBsAg level [14]. The APASL guidelines give similar recommendations but differ in that patients who are non-cirrhotic and have HBsAg levels of less than 10 IU/mL are excluded and placed in the low-risk category [26].

The AGA places patients who are receiving prednisone therapy for four or more weeks at a dose of more than 10 mg (or the equivalent with use of other types) into the high risk category [14]. The APASL and EASL guidelines require the same duration of steroid therapy but differ in that they require a higher dose; 20 mg or more prednisone in the APASL guidelines and more than 20 mg in the EASL guidelines [25,26].

The AGA and APASL guidelines include patients receiving TACE therapy for HCC in this category. The EASL guidelines also include TACE therapy but also other forms of HCC therapy (e.g., radiotherapy, resection, ablation, or systemic therapy) [14,25,26].

The three guidelines differ greatly in their placement of immune checkpoint inhibitors into the different risk categories. The APASL guidelines include patients who are on certain immune checkpoint inhibitors (nivolumab, pembrolizumab, atezolizumab, and ipilimumab) in this high-risk category [26]. By comparison, the AGA guidelines generally place the patients on immune checkpoint inhibitors in the moderate risk category [14], whereas the EASL guidelines place patients on those medications in the low risk group. Their reasoning is that these medications have the potential to strengthen the immune response towards HBV and that their perceived risk might be due to their frequent use alongside steroids and other cancer medications that increase the risk of reactivation [25].

The APASL and EASL guidelines also add patients receiving allogeneic or autologous hematopoietic stem cell transplantation to this category [25,26].

While the AGA guidelines place all patients receiving cytokine and integrin inhibitors in this group, the APASL and EASL guidelines separately address some medications in this class and place some of them in other risk groups. The EASL guidelines place patients on IL-17 and IL-6 inhibitors in this high-risk category [14,25,26].

The EASL guidelines specifically mention cyclophosphamide and high-dose combination chemotherapy such R-CHOP in this high-risk group [25].

##### HBsAg Negative/Anti-HBc Positive Patients (in the EASL Guidelines, These Group Recommendations Apply Only to Patients Who Are Also HBV DNA Negative)

As mentioned above, in the AGA, APASL, and EASL guidelines; patients are considered high-risk if they are on B cell depleting agents (e.g., anti-CD20 monoclonal antibodies) [14,25,26].

In the APASL guidelines, patients who are undergoing or planning to undergo allogeneic hematopoietic stem cell transplantation are also considered high risk, but the same does not apply for autologous hematopoietic stem cell transplantation recipients [26]. The EASL guidelines differ from them in that all patients receiving stem cell transplantation are placed in this high risk group [25].

The EASL guidelines include patients receiving TACE for HCC therapy, high dose combination chemotherapy (e.g., R-CHOP), anthracyclines, and T cell depleting therapy belatacept (when used in transplantation) in this group [25].

#### 5.3.2. Moderate Risk Category

##### HBsAg Positive Patients (in the EASL Guidelines Also Applies to Patients That Are HBsAg Negative/Anti-HBc Positive with Positive HBV DNA)

In the AGA and EASL guidelines, this category includes patients who are on anti-T cell therapy (for example abatacept) [14,25]. The APASL guidelines did not address this form of therapy due the lack of evidence at that time [26].

Patients on immune checkpoint inhibitors are also considered moderate risk according to the AGA guidelines [14]. This differs from the APASL and EASL guidelines which place those patients in the high-risk and low-risk groups, respectively [25,26].

The APASL guidelines also place patients who are undergoing cytotoxic chemotherapy, except for anthracyclines, in this risk category. In the EASL guidelines, the patients on anthracyclines, cyclophosphamide, and high dose combination chemotherapy (e.g., R-CHOP) are placed in the high-risk group, as stated above [25].

In the APASL guidelines, out of the anti-TNF agents, they make an exception for etanercept because it has lower potency and consider the patients on it as having moderate risk for reactivation, instead of high risk like the other anti-TNF agents. The AGA and EASL guidelines differ from the APASL guidelines in that they classify all the patients receiving anti-TNF agents as high risk [14,25,26].

The EASL guidelines place patients on anti-IL-12/23 (e.g., Ustekinumab) in the moderate risk group [25].

The APASL guidelines also consider the patients on bortezomib (a proteasome inhibitor) to be at moderate risk of reactivation [26]. This class of medications is not addressed in both the AGA and EASL guidelines [14,25].

The EASL guideline add the patients on mTOR inhibitors to this group. The AGA and APASL guidelines did not address them [14,26].

In the AGA guidelines, this category also includes patients on a prednisone dose of less than 10 mg (or the equivalent) for four or more weeks [14]. The APASL guidelines require a higher dose of 10–20 mg of prednisone (or the equivalent) for the same duration and consider patients on doses lower than that as low risk [26]. The EASL guidelines did not mention specific recommendations for the 10–20 mg dose of prednisone [25].

##### HBsAg Negative/Anti-HBc Positive Patients (in the EASL Guidelines, These Group Recommendations Apply Only to Patients Who Are Also HBV DNA Negative)

In the AGA guidelines, this group includes patients who are on anti-T cell therapy, immune checkpoint inhibitors, anthracyclines, anti-TNF agents, tyrosine kinase inhibitors, cytokine or integrin inhibitors, CAR-T cell therapy, anti-interleukin 6 agents, or JAK inhibitors. Notably, in the AGA guidelines, patients on immune checkpoint inhibitors are considered moderate risk whether they are HBsAg positive or HBsAg negative/anti-HBc antibodies positive [14]. The EASL guidelines mostly agree with the AGA except that they do not have specific recommendations for the patients who are on anti-IL6 or immune checkpoint inhibitors. They have separate recommendations for some cytokine or integrin inhibitors and specifically classify patients on anti-IL 12/23 and anti-IL 17 in this risk group [25]. They also differ from the AGA by placing the patients on anthracyclines therapy in the high-risk group. The APASL guidelines agree with placing the patients on anti-TNF agents in this group; however, they make a distinction for etanercept, since it has lower potency, and place patients that are on it in the low-risk category. APASL does not offer generalized recommendations for cytokine or integrin inhibitors. Their guidelines also do not offer recommendations for CAR-T cell therapy, anti-T cell therapy, and anti-IL6 therapy, and they place those receiving tyrosine kinase inhibitors in the low-risk group. The APASL guidelines, compared to the AGA guidelines, do not offer specific recommendations for patients that are HBsAg negative/anti-HBc positive and on immune checkpoint inhibitors [26].

According to the AGA guidelines, the patients included in this moderate-risk group also include those undergoing TACE therapy and those on 10 mg or more of prednisone (or the equivalent) for four or more weeks [14]. In the APASL guidelines, the only HBsAg negative/anti-HBc antibodies positive patients on steroids placed in any of the risk groups are those on 20 mg or more of steroid therapy and are placed in the low-risk group [26]. The EASL guidelines differ in that they place the patients (that are also HBV DNA negative) on more than 40 mg of prednisone (or the equivalent) in this moderate risk group. Those guidelines also add patients on cyclophosphamide to this group [25].

The APASL guidelines include the patients undergoing autologous hematopoietic stem cell transplantation in this category. As mentioned above, the EASL guidelines do not differentiate autologous from homologous stem cell transplantation and consider stem cell transplantation as high risk regardless of the serology subset. The APASL guidelines also include, similarly to the HBsAg positive patients, HBsAg negative/anti-HBc positive patients on Bortezomib (a proteasome inhibitor) in this group [25,26].

#### 5.3.3. Low Risk Category

##### HBsAg Positive Patients (in the EASL Guidelines, This Also Applies to Patients That Are HBsAg Negative/Anti-HBc Positive with Positive HBV DNA)

In this category, the AGA and APASL guidelines include patients that are on methotrexate and azathioprine therapy. The EASL guidelines agree with this and add mycophenolate mofetil. The AGA guidelines also include patients on mercaptopurine in this group [14,25,26].

The AGA guidelines also consider patients receiving intra-articular steroid injections, or those on prednisone (or the equivalent) at any steroid dose for a duration of a week or less to be low risk patients [14]. The APASL and EASL guidelines include those on a corticosteroid dose of less than 10 mg in this group [25,26].

The APASL guidelines include HBsAg positive patient with hepatitis C co-infection who are non-cirrhotic, receiving DAA therapy, and have HBsAg level of less than 10 IU/mL in this low-risk group [26]. In comparison, the AGA guidelines, as mentioned above, consider all HBsAg patients with HCV co-infection who receive DAA to be at high risk regardless of the degree of liver fibrosis or HBsAg level [14].

The EASL guidelines include patients on immune checkpoint inhibitors in this risk group. As mentioned above, the AGA and APASL guidelines place them in other risk groups [14,25,26].

##### HBsAg Negative/Anti-HBc Positive Patients (in the EASL Guidelines, These Group Recommendations Apply Only to Patients Who Are Also HBV DNA Negative)

According to the AGA guidelines, patients who are on methotrexate, 6-mercaptopurine, azathioprine, anti-TNF, and immune checkpoint inhibitor therapy are considered at a low risk for HBVr [14]. The EASL guidelines similarly place methotrexate and azathioprine in this group and add mycophenolate mofetil [25]. The APASL guidelines do not mention patients on methotrexate, 6-mercaptopurine, or azathioprine in this category. Both the EASL and APASL guidelines did not give recommendations for immune checkpoint inhibitors in this subpopulation, with APASL citing lack of sufficient evidence as the reason. Out of the anti-TNF agents, the APASL guidelines include patients on etanercept in this group. Those on other anti-TNF agents are placed in the moderate risk group, as mentioned above [25,26].

APASL guidelines also include HBsAg negative/anti-HBc antibodies-positive patients on cytotoxic chemotherapy, except for anthracyclines, in this group [25,26].

In the AGA guidelines, patients who are receiving intra-articular steroids, are on less than 10 mg of prednisone (or the equivalent) for four or more weeks, or who are on any dose of steroids for a week or less are considered low risk [14]. The APASL guidelines include patients on 20 mg or more of prednisone in this group [26]. The EASL guidelines include patients on less than 40 mg of prednisone for one week or less in this group [25].

As mentioned above, the APASL guidelines include the patients receiving tyrosine kinase inhibitors in this risk group, which contrasts with the AGA and EASL guidelines that place them in the moderate risk group [14,25,26].

Patients who have HCV coinfection and are undergoing DAA therapy are considered low risk, according to both the AGA and APASL guidelines [14,26]. This situation is not addressed in the EASL guidelines [25].

EASL guidelines include patients on mTOR inhibitors in this group. The AGA and APASL guidelines do not specifically address them [14,25].

### 5.4. Prophylaxis Therapy

The spectrum of disease of HBVr ranges from an asymptomatic illness to acute hepatitis to fulminant liver failure and death [28]. Without HBVr prophylaxis, 10% of patients will develop jaundice and 6% will have fulminant liver failure. Liver related death occurred in 5% of patients. HBVr also resulted in delay or interruption in chemotherapy, which resulted in increased cancer related mortality [29].

Nucleos(t)ide analogues (NAs) are used when antiviral prophylaxis is required. Lamivudine was used in the past, but it is no longer favored due to its low barrier to resistance. Entecavir (ETV), TDF, and TAF are preferred because of their higher barrier to resistance [25]. A meta-analysis published in 2016 evaluated the benefits of HBVr prophylaxis in patients receiving chemotherapy. It showed that prophylaxis with Tenofovir was the most effective in preventing HBVr and Entecavir decreased liver related risks the most between other NAs [30].

When given prophylaxis with NAs, the amount of reduction in the risk of reactivation is less in the moderate-risk group compared to the high-risk group, but it is still considerable to justify prophylaxis in certain scenarios [14]. In contrast, the risk reduction is less when comparing the moderate and low risk groups who are given the prophylactic antiviral therapy [14].

The APASL guidelines factor the presence of liver fibrosis in their prophylaxis recommendation and recommend testing for it. Their rationale is that patients with advanced fibrosis or cirrhosis have higher risk if reactivation occurs [26]. The AGA and EASL guidelines do not include the degree of liver fibrosis in the prophylaxis therapy decision-making process.

The APASL, AGA, and EASL guidelines agree that the high-risk group patients that are either HBsAg positive or HBsAg negative/anti-HBc antibodies positive should be prophylactically started on high barrier to resistance NAs [14,25,26]. See Table 2 and Figure 1.

For moderate-risk HBsAg positive patients, the APASL and EASL guidelines recommend starting prophylactic therapy [25,26]. The AGA guidelines differ in that they suggest starting the therapy, but they think it is sensible to monitor for reactivation if that is the patient’s preference and if there is low concern for poor compliance during monitoring. If monitoring is decided, they recommend testing for hepatitis B viral DNA and ALT every one to three months [14].

For moderate-risk and HBsAg negative/anti-HBc positive patients, the EASL guidelines advise close monitoring if HBV DNA is negative and the monitoring is conducted reliably. However, if the HBV DNA is positive their recommendation is starting with NAs [25]. The AGA guidelines suggest prophylaxis therapy, but they think it is reasonable to monitor for reactivation (more so than in HBsAg positive patients). The APASL guidelines consider the amount of liver fibrosis and recommend starting those with advanced fibrosis or cirrhosis on prophylactic therapy. Those without advanced fibrosis or cirrhosis can be monitored by measuring of ALT levels every 3 months. Once the level is more than 2 times the upper limit of normal, HBsAg and HBV DNA should be measured, and high barrier to resistance NAs should be started if either one of them is positive [14,26].

For low-risk patients, and for both those who are HBsAg positive or HBsAg negative/anti-HBc antibodies positive, the AGA guidelines suggest monitoring for reactivation. However, they find it acceptable to start prophylaxis therapy for patients who are risk-averse, do not place emphasis on the cost or demands of NAs therapy, and are on more than one low-risk IST. The APASL guidelines recommend checking for the degree of liver fibrosis and starting all patients with advanced fibrosis or cirrhosis on prophylactic therapy. Patients without advanced fibrosis or cirrhosis should be monitored by measuring ALT levels every 3 months [14,26]. The EASL guidelines recommend close monitoring for HBsAg negative/anti-HBc positive low-risk patients whether they are HBV DNA positive or negative [25].

The ASCO guidelines recommend starting antiviral prophylaxis for all HBsAg positive patients who are planned to undergo or are undergoing anticancer therapy, except for hormonal anticancer therapy. For the HBsAg negative/anti-HBc positive patients, they limit the start of antiviral prophylaxis therapy to those planned to be on anti-CD20 therapy or to undergo stem cell transplantation. The HBsAg negative/anti-HBc positive patient undergoing other types of anticancer therapy are monitored by checking HBsAg and ALT every 3 months during therapy. If they convert to being HBsAg positive, then high barrier to resistance NAs are started. If ALT rises then HBV DNA is checked and if it is more than 1000 IU/mL then NAs are started [31].

### 5.5. Duration of Prophylaxis Therapy

The AGA guidelines recommend continuing NAs for at least six months after stopping any IST. They recommend a longer duration of at least twelve months for B cell depleting medications. For HCV co-infected patients on DAA, they find it sensible to extend therapy to 6–12 months after cessation of the DAA therapy [14]. The APASL guidelines also advise considering discontinuation of NAs after six months of stopping IST for HBsAg negative/anti-HBC antibodies positive patients if they do not undergo seroconversion. However, they differ from the AGA guidelines in that for the HBsAg positive patients, they limit consideration of stopping NAs after 6 months only to patients without advanced fibrosis and with HBV DNA level that is less than 2000 IU/mL at baseline [26]. EASL recommend continuing NA therapy for at least 6–12 months after stopping IST. However, they recommend continuing therapy for at least 18 months in the case of use of B cell depleting agents [25]. The ASCO guidelines recommend continuing the high barrier to resistance NUCS in both HBsAg positive or HBsAg negative/anti-HBc positive patients for a minimum of 12 months starting from the completion of the anticancer therapy [31].

### 5.6. Cost-Effectiveness of Prophylaxis Therapy

Cost-effectiveness data for HBV prophylaxis during immunosuppression remain sparse, especially in the U.S., where direct studies are lacking. According to the AGA 2025 guideline, local drug costs and individual patient risk profiles are key determinants. The cost effectiveness of prophylaxis scales with baseline risk for HBV reactivation: it is clearly cost-effective for high-risk patients, potentially beneficial for those at moderate risk, and generally may not justified for low-risk individuals [14,25,26,33]. Entecavir (ETV), tenofovir disoproxil fumarate (TDF), and tenofovir alafenamide (TAF) are all highly effective in preventing HBV reactivation during immunosuppression and all are considered to be cost-effective options for HBVr high risk patients [34,35,36,37]. When selecting among these agents, safety considerations, particularly renal and bone toxicity, and insurance coverage remain the primary factors directing clinical decision-making [38].

## 6. HBV Reactivation Treatment

The main goal of HBVr treatment is to stop viral replication and consequently restore the immune system regulation [19].

Regardless of the presence or absence of hepatitis flare (in other words, elevation of ALT levels), all immunosuppressed patients with evidence of HBVr should be treated with NAs. The treatment should be initiated as soon as possible, because 25–50% of patient with HBVr can develop severe hepatitis and liver failure [39].

While NAs and PEG-IFNα are used in the treatment of chronic HBV infection [25], only NAs are recommended for HBVr treatment and prophylaxis [25,26]. NAs inhibit HBV polymerase which in turn inhibits DNA synthesis and viral replication. They also have immunomodulatory properties. By decreasing the viral load and therefore the exposure to viral antigens, NAs reduce the activation of the immune system and subsequently the associated hepatic inflammation secondary to the immune activation. They also increase the activity of immune cells, which enhances the anti-viral response [19]. The NAs are divided based on barrier to HBV resistance into low and high barrier to resistance. The low-barrier to resistance NAs are Lamivudine and Adefovir. The high-barrier to resistance NAs are Entecavir and Tenofovir [40]. The AGA, EASL, and APASL guidelines all recommend the use of NAs with high barrier to resistance [14,25,26].

## 7. Conclusions

HBV infection is a global health concern. With the increasing utilization of immunosuppressive therapy, HBVr significance has grown in clinical practice. The most recent clinical guidelines on HBVr are provided by three major societies and include the AGA 2025, EASL 2025, and APASL 2021 guidelines.

This narrative review focuses in comparing these three current guidelines, highlighting key similarities and differences to guide clinicians in their decisions regarding the utilization of antivirals.

Universal HBV screening, already recommended by WHO for all adults aged 18 years or older [2], is similarly required in all patients at risk of immunosuppression through the use of HBsAg and total anti-HBc.

The risk of HBVr during IST has been stratified into three categories in all the three guidelines: high (>10%), moderate (1–10%), and low (<1%). Some differences in risk stratification exist between societies, mainly between the 1–10% and the >10% risk groups. The effectiveness of prophylaxis is directly related to the baseline risk for HBVr.

In all the three guidelines, prophylaxis is clearly indicated for all of the high-risk groups. Prophylaxis for these groups yields the greatest absolute reduction in HBVr and thus is cost-effective.

Although the benefit is not as clear-cut as in the high-risk group, prophylaxis for the moderate-risk group remains justifiable. In contrast, prophylaxis for low-risk individuals generally is usually unnecessary.

The APASL 2021 guidelines differ from the other guidelines in that the presence of advanced liver fibrosis or cirrhosis in itself is regarded as an indication for prophylaxis across all risk groups, including the moderate and low risk ones.

The AGA 2025 and EASL 2025 are largely in agreement in their recommendations regarding the moderate risk group and both favor close monitoring. An exception is noted for HBsAg(−)/anti-HBc(+) positive patients that do not have detectable HBV DNA for whom EASL recommends monitoring closely, but AGA suggests prophylaxis therapy regardless of HBV DNA status.

Entecavir, tenofovir disoproxil fumarate, and tenofovir alafenamide are all highly effective in preventing HBVr during immunosuppression, and all are considered to be cost effective options. When selecting among these agents, safety considerations, particularly renal and bone toxicity, and insurance coverage remain the primary factors dictating clinical decision making.

It is clear that the guidelines do not fully agree in all their recommendations. This is partly due to the lack of strong evidence that they based their recommendations on. As more evidence accumulates, this will guide future guidelines better. For now, when the guidelines disagree in their recommendations, we favor selection of the more cautious recommendation. However, the discrepancy in the recommendations has a positive aspect in that it gives flexibility to the managing physician, especially in difficult circumstances such as low resource areas.

Although these clinical guidelines offer a structured approach to patient care, clinical judgment and patient involvement in the management plan are still essential.

In the future, it might be helpful for the major societies to unify their recommendations and release a consensus guideline which can be updated regularly as new evidence becomes available.

## Figures and Tables

**Figure 1 diseases-13-00355-f001:**
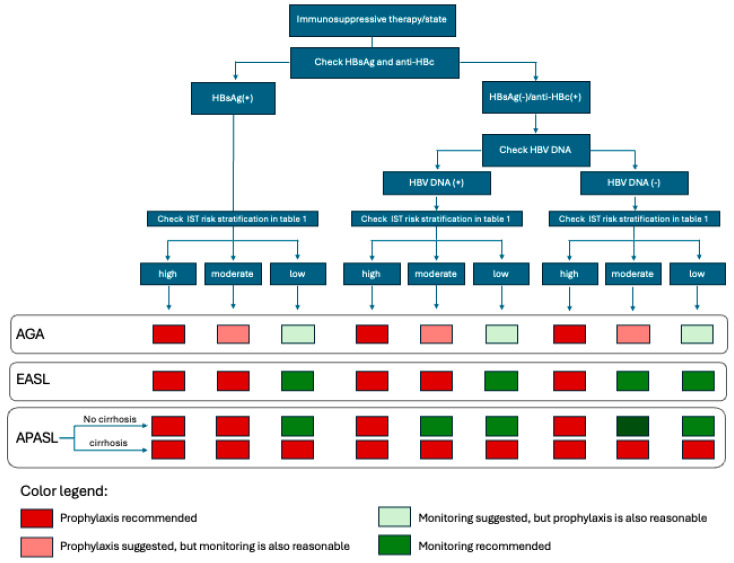
Comparison of the indications for initiation of antiviral prophylaxis between the 2025 AGA, 2025 EASL, and 2021 APASL guidelines.

**Table 1 diseases-13-00355-t001:** Differences in the therapy HBVr risk stratification between the major gastroenterology/liver society latest guidelines.

Therapy/Medication	Serology	AGA	APASL	EASL
**B Cell depleting agents**	HBsAg(+)	High risk	High risk ^a^	High risk
	HBsAg(−)/anti-HBc(+)	High risk	High risk ^a^	HBV DNA(+): High risk HBV DNA(−): High risk
**Anti-TNF therapy**	HBsAg(+)	High risk	High risk ^b^	High
	HBsAg(−)/anti-HBc(+)	Low risk	Moderate risk ^b^	HBV DNA(+): High risk HBV DNA(−): Moderate risk
**CAR-T cell therapy**	HBsAg(+)	High risk	Not addressed	High risk
	HBsAg(−)/anti-HBc(+)	Moderate risk	Not addressed	HBV DNA(+): High risk HBV DNA(−): Moderate risk
**Chemotherapy**	HBsAg(+)	Anthracyclines addressed and considered high risk	Moderate risk if cytotoxic chemotherapy, except anthracyclines, which are high risk	High risk if high dose combination chemotherapy or anthracyclines
	HBsAg(−)/anti-HBc(+)	Anthracyclines addressed and considered moderate risk	Low risk if cytotoxic chemotherapy, except anthracyclines and proteasome inhibitors, which are moderate risk	HBV DNA(+): High risk if high dose combination chemotherapy or anthracyclines HBV DNA(−): High risk if high dose combination chemotherapy or anthracyclines
**Cytokine/integrin inhibitors (includes anti-IL-6 therapy)**	HBsAg(+)	High risk	APASL was uncertain about Natalizumab and Tocilizumab. Ustekinumab is moderate risk. Other drugs in this class were not addressed	anti-IL-12/23 is moderate risk. Anti-IL-17 and IL-6 are high risk. The others are not addressed.
	HBsAg(−)/anti-HBc(+)	Moderate risk	Ustekinumab is moderate risk. Other drugs in this class were not addressed	HBV DNA(+): Anti-IL-12/23 are moderate risk. Anti-IL-17 and IL-6 are high risk. HBV DNA(−): Anti-IL-17 and IL-12/23 are moderate risk. Anti-IL-6 are not addressed
**TACE**	HBsAg(+)	High risk	High risk	High risk (also radiotherapy, resection, ablation, and systemic therapies)
	HBsAg(−)/anti-HBc(+)	Moderate risk	Not addressed	HBV DNA(+): High risk (also radiotherapy, resection, ablation, and systemic therapies) HBV DNA(−): High risk in case of TACE therapy.
**Tyrosine kinase inhibitor therapy**	HBsAg(+)	High risk	High risk ^c^	High risk
	HBsAg(−)/anti-HBc(+)	Moderate risk	Low risk	HBV DNA(+): High risk HBV DNA(−): Moderate risk
**JAK inhibitor therapy**	HBsAg(+)	High risk	Not addressed	High risk
	HBsAg(−)/anti-HBc(+)	Moderate risk	Not addressed	HBV DNA(+): High risk HBV DNA(−): Moderate risk
**DAA therapy for HCV infection**	HBsAg(+)	High risk	High risk but low risk for non-cirrhotic patients with HBsAg < 10 IU/ml	Not addressed
	HBsAg(−)/anti-HBc(+)	Low risk	Low risk	Not addressed
**Anti-T cell therapy**	HBsAg(+)	Moderate risk	Abatacept mentioned as having an uncertain risk	Moderate risk
	HBsAg(−)/anti-HBc(+)	Moderate risk	Not addressed	HBV DNA(+): Moderate risk HBV DNA(−): High risk if belatacept (in the setting of transplantation) and moderate risk if abatacept
**Immune check point inhibitors**	HBsAg(+)	Moderate risk	High risk	Low risk
	HBsAg(−)/anti-HBc(+)	Low risk	Uncertain risk	HBV DNA(+): Low risk HBV DNA(−): Not addressed
**Azathioprine, methotrexate and mycophenolate mofetil**	HBsAg(+)	Low risk ^d^	Low risk ^d^	Low risk
	HBsAg(−)/anti-HBc(+)	Low risk ^d^	Not addressed	Low risk
**Cyclophosphamide**	HBsAg(+)	Not addressed	Not addressed	High risk
	HBsAg(−)/anti-HBc(+)	Not addressed	Not addressed	HBV DNA(+): High risk HBV DNA(−): Moderate risk
**mTOR inhibitors**	HBsAg(+)	Not addressed	Not addressed	Moderate risk
	HBsAg(−)/anti-HBc(+)	Not addressed	Not addressed	HBV DNA(+): Moderate risk HBV DNA(−): Low risk
**Stem cell transplantation**	HBsAg(+)	Not addressed	High if hematopoietic stem cell transplantation	High risk
	HBsAg(−)/anti-HBc(+)	Not addressed	High if allogeneic HSCT. Moderate if autologous HSCT.	High risk
**Steroids therapy**	HBsAg(+)	High risk if prednisone dose is ≥10 mg/day ≥ 4 weeks. Moderate risk if the prednisone dose is <10 mg/day ≥ 4 weeks. Low risk at any prednisone dose if for ≤1 week ^e^	High risk if prednisone dose is ≥20 mg for ≥4 weeks. Moderate risk if prednisone dose is 10–20 mg/day for ≥4 weeks. Low risk if prednisone dose is <10 mg/day.	High risk if the corticosteroid dose is >20 mg/day for > 4 weeks. Low risk if the corticosteroids dose is <10 mg/day.
	HBsAg(−)/anti-HBc(+)	Moderate risk if prednisone dose is ≥10 mg/day for ≥4 weeks. Low risk if prednisone dose is <10 mg/day for ≥4 weeks. Low risk at any dose of prednisone if for ≤1 week ^e^	Low risk if ≥20 mg prednisone dose.	HBV DNA(+): High risk if the corticosteroid dose is >20 mg for >4 weeks. Low risk if the corticosteroids dose is <10 mg/day. HBV DNA(−): Moderate risk if the corticosteroids dose is >40 mg/day. Low risk if the corticosteroids dose is <40 mg/day for ≤1 week.

Note. HBsAg(+) = Hepatitis B surface antigen positive; HBsAg(−)/anti-HBc(+) = Hepatitis B surface antigen negative and hepatitis B core antibody positive; anti-TNF = anti-tumor necrosis factor; CAR-T = chimeric antigen receptor T cell; IL = interleukin; TACE = transcatheter arterial chemoembolization; JAK = Janus Kinase; DAA = direct-acting antiviral; HCV = hepatitis C virus; mTOR = mammalian target of rapamycin. ^a^: APASL was uncertain about Ocrelizumab and Ibritumomab. ^b^: APASL considers lower potency anti-TNF (etanercept) as moderate risk if HBsAg(+) and low risk if HBsAg(−)/anti-HBc(+). ^c^: APASL was uncertain about Ibrutinib. ^d^: AGA and APASL did not address mycophenolate mofetil. ^e^: AGA considers intra-articular steroids therapy as low risk in all serology groups. The other guidelines do not address this form of therapy.

**Table 2 diseases-13-00355-t002:** Comparison between the latest major gastroenterology/liver society guidelines regarding initiating antiviral prophylaxis in different reactivation risk groups.

Risk of Reactivation	AGA 2025 Guidelines	APASL 2021 Guidelines	EASL 2025 Guidelines
**High > 10%**			
**HBsAg(+)**	Prophylaxis recommended	Prophylaxis recommended	Prophylaxis recommended
**HBsAg(−)/anti-HBc(+)**	Prophylaxis recommended	Prophylaxis recommended	Prophylaxis recommended
**Moderate 1–10%**			
**HBsAg(+)**	Suggest prophylaxis but monitoring is also reasonable	Prophylaxis recommended	Prophylaxis recommended
**HBsAg(−)/anti-HBc(+)**	Suggest prophylaxis but monitoring is also reasonable	Prophylaxis recommended if advanced fibrosis or cirrhosis. Otherwise, monitoring is recommended.	Prophylaxis recommended if HBV DNA is positive. Close monitoring if HBV DNA is negative
**Low < 1%**			
**HBsAg(+)**	Suggest monitoring but prophylaxis is acceptable	Prophylaxis recommended if advanced fibrosis or cirrhosis. Otherwise, monitoring is recommended.	Monitor closely
**HBsAg(−)/anti-HBc(+)**	Suggest monitoring but prophylaxis is acceptable	Prophylaxis recommended if advanced fibrosis or cirrhosis. Otherwise, monitoring is recommended.	Monitor closely

## Data Availability

Data sharing is not applicable.

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
