# Peer review of "Navigating the Latest Hepatitis B Virus Reactivation Guidelines"

_diseases, 2025, doi:10.3390/diseases13110355_

Round 1

Reviewer 1 Report

Comments and Suggestions for Authors
  1. This review is quite comprehensive, considering the evaluation of reactivation in HBV-infected patients receiving immunosuppressive therapy using three different guidelines. It reveals the different and common perspectives of the guidelines in a single review. I believe it will fill a gap in the field in these respects.
  2.  A detailed comparison of the guidelines on this topic in the table will greatly facilitate the reader's practical approach. The results, the evidence presented, and the arguments are consistent with the purpose of the review.
  3.  The tables are sufficient and descriptive.
  4.  It would be better if the references were strengthened with up-to-date sources.

Author Response

Comments 1: This review is quite comprehensive, considering the evaluation of reactivation in HBV-infected patients receiving immunosuppressive therapy using three different guidelines. It reveals the different and common perspectives of the guidelines in a single review. I believe it will fill a gap in the field in these respects.

Response 1: We appreciate the reviewer's comment.

Comments 2: A detailed comparison of the guidelines on this topic in the table will greatly facilitate the reader's practical approach. The results, the evidence presented, and the arguments are consistent with the purpose of the review.

Response 2: We appreciate the reviewer's comment.

Comments 3: The tables are sufficient and descriptive.

Response 3: We appreciate the reviewer's comment.

Comments 4: It would be better if the references were strengthened with up-to-date sources.

Response 4: We added several new and up-to-date references (i.e., published after 2020), especially on recent advances in understanding the pathophysiology of Hepatitis B reactivation.

Some of the references added: 

(Liu C, Shih YF, Liu CJ. Immunopathogenesis of Acute Flare of Chronic Hepatitis B: With Emphasis on the Role of
Cytokines and Chemokines. Int J Mol Sci. 2022;23)

(Li M, Gao Y, Yang L, Lin Y, Deng W, Jiang T, Bi X, Lu Y, Zhang L, Shen G, Liu R, Wu S,
Chang M, Xu M, Hu L, Song R, Jiang Y, Yi W, Xie Y. Dynamic changes of cytokine profiles and virological
markers during 48 weeks of entecavir treatment for HBeAg-positive chronic hepatitis B. Front Immunol.
2022;13:1024333)

(Jin X, Bi J. Prospects for NK-based immunotherapy of chronic HBV infection. Front Immunol. 2022;13:1084109).

(Ma H, Yan QZ, Ma JR, Li DF, Yang JL. Overview of the
immunological mechanisms in hepatitis B virus reactivation: Implications for disease progression and
management strategies. World J Gastroenterol. 2024 Mar 14;30(10):1295-1312)

(Chen S, Li B, Luo W, Rehman AU, He M, Yang Q, Wang S, Guo J, Chen L, Li X. Paclitaxel-induced Immune
Dysfunction and Activation of Transcription Factor AP-1 Facilitate Hepatitis B Virus Replication. J Clin Transl
Hepatol. 2024 May 28;12(5):457-468)

Reviewer 2 Report

Comments and Suggestions for Authors

The review by Zeyad Elharabi et al. is a narrative review comparing the current international guidelines (APASL, EASL, and AGA) about hepatitis B virus reactivation during or after immunosuppressive therapies.

Authors should include information about their review method and improve the information and tables before being considered for publication in Diseases.

General comment

  1. Abstract
  • Please include specific details about the study’s methodology to elaborate on this narrative review
  1. Introduction
  • The introduction provides a good overview but could benefit from more relevant literature. Specifically, to briefly discuss recent advancements in understanding the hepatitis B reactivation mechanisms. Please incorporate more references to recent studies published between 2020 and 2025
  1. Material and Methods
    • Please include a specific paragraph describing the study’s methodology to elaborate on this narrative review
  2. Tables
  • The size of Table 1 is excessive. Please try to summarize the table or split their information. Please, include a list of abbreviations for a better understanding
  1. Conclusion
    • Please include a paragraph summarizing the key findings of this review, the study’s contribution to the field, and its potential impact on future research or clinical guidelines. Please highlight the study’s contribution and its implications for clinical practice.

Author Response

Thank you for your comments. They are greatly appreciated. 

Comments 1: Authors should include information about their review method and improve the information and tables before being considered for publication in Diseases.

Response 1: We have now clearly mentioned in the abstract in page 1 and in a newly added methods section in page 3 that the review by Zeyad Elharabi et al. is a narrative review. We shortened table 1 and tried to improve the content according to the comments from the reviewers. Please see below.

Comments 2: Abstract:

Please include specific details about the study’s methodology to elaborate on this narrative review

Response 2: We have now clearly mentioned in the abstract and methods sections that the review by Zeyad Elharabi et al. is a narrative review. We added a methods section in page 3 and we gave a more detailed definition of the narrative review in this section.

Comments 3: Introduction

The introduction provides a good overview but could benefit from more relevant literature. Specifically, to briefly discuss recent advancements in understanding the hepatitis B reactivation mechanisms. Please incorporate more references to recent studies published between 2020 and 2025

Response 3: Thank you for the suggestion. We added two new paragraphs in the introduction section in page 2 to briefly discuss recent advancements in understanding the hepatitis B reactivation mechanisms and added a total of 5 recent publications with publication years being between 2022 to 2024. One of the publications was from 2019.

Comments 4: Material and Methods

Please include a specific paragraph describing the study’s methodology to elaborate on this narrative review

Response 4:  We added a specific section for materials and methods in page 3 to detail the narrative review methodology

Comments 5: Tables

a)The size of Table 1 is excessive. Please try to summarize the table or split their information.

b)Please, include a list of abbreviations for a better understanding

Response 5:

  1. a) We shortened and reorganized Table 1 so that it is more compact and reduced the size without losing content. We also removed notes the column and adding the information there as a footnote to the table.
  2. b) We added abbreviations below Table 1. Please note that we also have an abbreviation list in page 16. This list was updated with new abbreviations after new content was added in response to comments received.

Comments 6: Conclusion

Please include a paragraph summarizing the key findings of this review, the study’s contribution to the field, and its potential impact on future research or clinical guidelines. Please highlight the study’s contribution and its implications for clinical practice.

Response 6: A conclusion section was added in page 15 addressing the points in the comment.

Reviewer 3 Report

Comments and Suggestions for Authors

General Evaluation

This is a comprehensive and timely review comparing the latest guidelines (AGA 2025, EASL 2025, APASL 2021) on HBV reactivation in patients receiving immunosuppressive therapy. The topic is highly relevant given the rising use of novel immunomodulators and the lack of agreement between guidelines. The manuscript is well-structured and includes valuable comparative tables. However, the paper is largely descriptive and would benefit from more critical synthesis, clearer language, and stylistic refinement to improve clinical applicability.

Major Comments

  • The manuscript should clearly state whether this is a narrative review or a scoping review.
  • When summarizing the guidelines, the authors should explicitly discuss how clinicians should approach conflicting recommendations (e.g., immune checkpoint inhibitors classified as high risk by APASL, moderate by AGA, and low by EASL).
  • The comparative tables are helpful, but the addition of a flowchart/algorithm for clinical decision-making (screening, prophylaxis, monitoring) would greatly increase the practical utility of this review.
  • Some important areas are only briefly mentioned and the expansion on these aspects would enhance the manuscript’s relevance: cost-effectiveness and feasibility of different prophylaxis strategies; strategies in special populations (pregnancy, HIV co-infection..).
  • The discussion is currently descriptive. The authors should provide a stronger expert perspective, highlighting clinical take-home points and suggesting future directions for research and guidelines harmonization.

Minor Comments

Abstract

  • The abstract reads more like an introduction. It should include a concise summary of findings and practical implications.

Introduction

  • Avoid redundancy (e.g., “significant morbidity and mortality” appears several times).
  • Consider expanding on the global HBV burden, particularly in low- and middle-income countries.

Prophylaxis Section

  • Use “prophylactic therapy” instead of “the prophylaxis therapy.”
  • Highlight controversies more clearly, especially around moderate-risk groups.

Tables

  • The Tables format is dense, consider simplifying categories, if possible
  • Use consistent terminology (e.g., “HBsAg positive” vs. “HBsAg (+)”).
  • Move explanatory notes to footnotes for readability.

Summary

The manuscript addresses an important clinical issue and is a valuable contribution. However, I recommend major revision to:

  • Clarify the review type.
  • Provide more critical synthesis and expert guidance.
  • Add visual aids (flowchart/algorithm).

Author Response

Comments : This is a comprehensive and timely review comparing the latest guidelines (AGA 2025, EASL 2025, APASL 2021) on HBV reactivation in patients receiving immunosuppressive therapy. The topic is highly relevant given the rising use of novel immunomodulators and the lack of agreement between guidelines. The manuscript is well-structured and includes valuable comparative tables. However, the paper is largely descriptive and would benefit from more critical synthesis, clearer language, and stylistic refinement to improve clinical applicability.

We appreciate the very encouraging, positive comments of the reviewer. We tried to update the manuscript in line with the comments of the reviewer with the hope to fulfill the expectations.

Comments 1: The manuscript should clearly state whether this is a narrative review or a scoping review.

Response 1: We clearly state that that review is a narrative one in both the abstract in page 1 and the newly added methods section in page 3.

Comments 2: When summarizing the guidelines, the authors should explicitly discuss how clinicians should approach conflicting recommendations (e.g., immune checkpoint inhibitors classified as high risk by APASL, moderate by AGA, and low by EASL).

Response 2: Explanation added to explain the big discrepancy in the recommendations for the immune checkpoint inhibitors in page 6 paragraph 8. The general reason behind the discrepancy between the guidelines was addressed in the newly added conclusion section in page 15.

Comments 3: The comparative tables are helpful, but the addition of a flowchart/algorithm for clinical decision-making (screening, prophylaxis, monitoring) would greatly increase the practical utility of this review.

Response 3: A flowchart was added (figure 1 in page 13) to make the prophylaxis versus monitoring decision making easier and for enabling the quick comparison between the addressed guidelines.

Comments 4: Some important areas are only briefly mentioned and the expansion on these aspects would enhance the manuscript’s relevance:

  1. a) cost-effectiveness and feasibility of different prophylaxis strategies;
  2. b) strategies in special populations (pregnancy, HIV co-infection).

Response 4:

  1. A section was added discussing the cost-effective of the different prophylaxis strategies in page 14 at the bottom of the page (5.6 Cost-effectiveness of prophylaxis therapy)
  2. Added mention of pregnancy and ESRD on the top of the already mentioned special populations in the introduction section (page 2, 3rd paragraph). Not each of the three guidelines address the reactivation in the mentioned special populations. Therefore, comparison among the guidelines is not applicable.

Comments 5: The discussion is currently descriptive. The authors should provide a stronger expert perspective, highlighting clinical take-home points and suggesting future directions for research and guidelines harmonization.

Response 5: This was addressed in the newly added conclusion section in page 15.

Comments 6: Abstract

The abstract reads more like an introduction. It should include a concise summary of findings and practical implications.

Response 6: The abstract in page 1 was completely revised to go in line with the reviewer’s comment.

Comments 7: Introduction

Avoid redundancy (e.g., “significant morbidity and mortality” appears several times).

Consider expanding on the global HBV burden, particularly in low- and middle-income countries.

Response 7: Stated redundancy was addressed and it is now only mentioned once.

Additions were added to the introduction to address the burden in the low and middle income countries (page 2, first paragraph).

Comments 8: Prophylaxis Section

Use “prophylactic therapy” instead of “the prophylaxis therapy.”

Highlight controversies more clearly, especially around moderate-risk groups.

Response 8: Prophylactic therapy changed to prophylaxis therapy.

Explanation for the discrepancy in the immune checkpoint inhibitors was added. The general reason behind the discrepancies was addressed in the conclusion section that was added in page 15.

Comments 1: Tables

The Tables format is dense, consider simplifying categories, if possible

Use consistent terminology (e.g., “HBsAg positive” vs. “HBsAg (+)”).

Move explanatory notes to footnotes for readability.

Response 1:

The categories were combined for better readability. The terminology was unified as suggested.

The explanatory notes were moved to the footnotes which we agree did ease the readability.

Comments 1: Summary

The manuscript addresses an important clinical issue and is a valuable contribution. However, I recommend major revision to:

Clarify the review type.

Provide more critical synthesis and expert guidance.

Add visual aids (flowchart/algorithm).

Response 1:

The review type was clarified in the abstract and method sections.

Added opinion in the conclusion section in page 15.

Added a flowchart as a visual aid in page 13.

Round 2

Reviewer 2 Report

Comments and Suggestions for Authors

The authors have significantly improved the quality of their narrative review by comparing current guidelines on hepatitis B virus reactivation after considering the reviewers' recommendations.